# Two-period e-commerce platforms operation strategies considering the difference in product quality perception

Bo Xie[1], Qiqi Guo[1], Yingying Cheng[1]*, Muqing Niu[2]

1 Business School and MBA Education Center of Henan University of Science and Technology, Henan, China, 2 Affiliated First Hospital, Henan University of Science and Technology, Henan, China

* 9906180@haust.edu.cn

## Abstract

The difference in consumer perceptions of product quality influences consumers' propensity to purchase through various channels, thereby affecting the profit of e-commerce platforms and manufacturers. This factor is a critical consideration for e-commerce platforms when formulating their operational strategies in a multi-channel competitive environment. This study focuses on a product retail supply chain comprising two e-commerce platforms, two manufacturers, and strategic consumers. Innovatively, we construct a two-period dynamic pricing game model to compare commission-based and self-operated models for the platforms and deeply explore the impact of consumers' product quality perception differences on pricing strategies and platform operation model choices. Four scenarios are considered: self-operated model under simultaneous decision-making, commission-based model under simultaneous decision-making, self-operated model under successive decision-making, and commission-based model under successive decision-making. Results show that as the probability of consumers perceiving high-quality products increases, both e-commerce platforms and manufacturers can enhance their profit by raising prices. Moreover, when platforms make decisions simultaneously, they should choose the commission-based model. In contrast, when platforms make decisions sequentially, the leader platform should choose the self-operated model, while the follower platform should choose the commission-based model. In particular, in the context of successive decision-making, after attracting consumers through the low-price strategy in the first period, the leader platform can still effectively maintain market share even if the price is raised in the second period.

## 1. Introduction

The transformation of e-commerce platforms from a single-channel to multi-channel coexistence has intensified channel competition [1,2], and consumers frequently

**Data availability statement:** All relevant data are within the paper and its Supporting Information files.

**Funding:** This research was funded by Soft science project of Henan Province(grant # 252400411280) awarded to BX, Soft science project of Henan Province (grant # 252400410046) awarded to YC, Philosophy and social science education strong provincial project of Henan Province (grant # 2025JYQS1079) awarded to BX, and Henan Provincial Postgraduate Course Ideological and Political Demonstration Course Project (grant # YJS2024SZ14) awarded to BX. The funders had no role in study design, data collection and analysis, decision to publish, or preparation of the manuscript.

**Competing interests:** The authors have declared that no competing interests exist.

switch purchasing channels during the multi-period buying process to seek the best choice [3]. For example, Walmart's Walmart+ Assist program and Amazon Prime compete for users with exclusive discounts [4], demonstrating the prevalence of consumer switching between different platforms in a multi-channel coexistence environment. Especially during large-scale promotional events such as "Double 11", the number of users and orders on Jingdong and Tmall increased significantly [5], highlighting consumers' behavior of searching for the best purchasing channels across multiple platforms. Consumer strategy-based purchasing behavior places higher demands on the operational strategies of e-commerce platforms.

It has been shown that during multiple product purchases, consumer buying behavior exhibits strategic consumption characteristics, where product quality [6], the price [7,8], service [9], and factors such as out-of-stock [10], utility [11], retailer-driven [12], and discount period cut-off [13] can be the determinants of consumers' switching channels, with the price factor playing a significant role. Strategic consumers tend to favor more competitively priced channels and seek discounts and promotions more aggressively [14,15]. Manufacturers and platforms adopt price discrimination strategies to enhance competitiveness [16]. E-commerce platforms can be categorized into self-operated platforms and commission-based platforms according to the different roles they play in sales [17]. Jingdong has created a self-operated brand called "Made in Jingdong", and at the same time allows third-party merchants to set up and operate their own shops [18]. Under the self-operated model, platforms attract consumers through price discounts. For example, Apple launched the "Apple Trade In" under the national subsidy policy in May 2025, which allows consumers to purchase iPhones from Jingdong with both national subsidies and trade-in subsidies [19]. Under the commission-based model, platforms attract consumers by issuing coupons. For instance, Tmall invested a total of 30 billion in consumer coupons during the "Double 11" period in 2024, which boosted consumers' incentives to shop on Tmall [20]. Through in-depth analysis of the behavioral patterns of target consumer groups, e-commerce platforms can accurately grasp their consumption psychology and purchasing decision-making process [21,22]. At the same time, the platforms develop operational strategies to attract and motivate consumers to make purchases, which in turn drives the platform's profit growth [23].

Currently, research on two-period pricing typically assumes that consumer valuations are fixed or that price is the only motivation for transfer [24,25]. Although some literature has considered consumers' strategic behavior, few have incorporated "perceived quality updates" into dynamic pricing models. Although product quality perception is widely recognized as a key factor influencing consumer decision-making [26,27], most studies focus on the formation mechanisms of quality perception in static environments (such as online reviews and brand awareness), focusing on product quality itself or price competition [8,9]. For example, improving objective product quality through supply chain optimization, quality certification, or technological upgrades [27,28] while ignoring its impact on platform operational strategies (such as pricing and commission models) in multi-cycle dynamic competition [28,29], and systematic exploration of how differences in consumer quality perception affect platform pricing and model selection at the strategic level remains insufficient.

Although there is a wealth of research on price discrimination and platform operating models [17,30], studies on the choice between platform self-operation and commission models have mainly focused on aspects such as channel power, commission rates, and showroom behavior [31,32]. Few scholars have combined research on consumer quality perception differences with dynamic two-period game analysis, and have not fully explored how consumers' channel switching behavior based on quality perception in the second period affects the platform's optimal pricing and model selection. Most of the current research is based on a single platform or static model [33], and there's a lack of theoretical modeling of dynamic interactions under dual-platform competition. Few scholars have done a systematic analysis of the interaction between quality perception differences and platform decision order.

In response to the shortcomings of existing research, we construct a two-period dynamic game model, which classifies strategic consumers into two types of high and low perception according to the different valuations of consumers' perception of product quality [24,34]. Considering the impact of price factors on channel competition in the context of differences in consumers' perception of product quality, we divide the enterprise strategy portfolio into four types: self-operated model under simultaneous decision-making by both platforms (S-R), commission-based model under simultaneous decision-making by both platforms (S-C), self-operated model under sequential decision-making by both platforms (D-R), and commission-based model under sequential decision-making by both platforms (D-C). Considering the uncertainty factors of product price and consumer product quality perception at different stages, we provide a new theoretical framework for studying the operational decisions of different types of platforms in different periods, which helps platforms and manufacturers to under-stand the market dynamics and consumer behaviors better so that they can flexibly adjust their pricing strategies, commission policies and incentives in actual operations. This paper clearly answers the following research questions: 1) the choice of pricing strategies of manufacturers and platforms under different operating modes. 2) The impact of commission rate and discount amount on decision-making under different decision sequences of enterprises under the same operating mode. 3) The impact of the probability changes of consumer product quality perception difference on the profit of manufacturers and platforms.

The rest of the paper is organized as follows. In Section 2, we systematically sort out the theoretical literature related to this study and establishes the theoretical base of the study; In Section 3, we construct the basic framework of the mathematical model, clarify the assumptions of the research problem and standardizes the model parameter notation system; In Section 4, we develop a theoretical framework encompassing four strategic combinations and derive their respective equilibrium solutions through game-theoretic analysis; In Section 5, we conduct a multi-dimensional comparative study, focusing on the analysis of the dynamic changes of product full-period pricing, channel incentive strategy and supply chain profit distribution under the self-management mode and commission mode, and reveal the role of quality perception difference and channel competition intensity on the platform's operation mode selection and the implementation effect of the price discrimination strategy; In Section 6, we use numerical simulation to verify the validity of the theoretical model, and through sensitivity analysis, explore the influence of key parameter changes on supply chain equilibrium decision-making and optimal profit, and provide model support and suggestions for the selection of optimal operation strategies for e-commerce platforms under different circumstances; In Section 7, we summarize the main findings and put forward the managerial implications. The equilibrium-solving process and proof of theorems in this paper are shown in the appendix.

## 2. Literature review

This study mainly involves three aspects: consumer purchase intention and product quality perception, price discrimination, and e-commerce platform operation strategy. This subsection will review the literature on these three aspects.

Consumers' perception of product quality is a key factor influencing their purchasing decisions. In early studies on consumers' two-period buying behavior, scholars argued that consumers' product preferences are fixed and that they switch purchasing channels solely due to price differences; the first purchase conveys the preference, and the only motivation for switching is the lower price [25,35,36]. Fudenberg et al. [25] constructed a two-period game model, proposing that

 

consumers' preferences are completely independent over time. Shin and Sudhir [37] make a modified formulation of preference randomness to adapt it to the nature of correlation over time. Subsequent scholars pointed out that consumer valuations in multi-cycle purchases are uncertain [38], in a two-period purchasing supply chain where there is multi-channel competition, consumers are unable to accurately assess the quality of the product in the first period [39] and obtain the actual value of the product they have purchased in the second period [40], and if consumers realize a low value in the first period, switching purchases in the second period implies a higher expected product value [41]. Numerous studies have shown that product quality perception [6], product prices [8,42], platform live streaming [42,43], and online product reviews [44,45] are the critical factors influencing consumers' purchasing decisions. Among them, consumers' perception of online product quality is a subjective comparison of the quality and value of different products under the combined effect of personal factors and shopping situations, which is between objective and subjective [26] and is influenced by internal and external information of enterprises [46]. Consumers' expected valuation of product quality refers to consumers' prior beliefs about product quality, which can be divided into high prior quality and low prior quality [45]. Consumers' perception of product quality in online shopping is affected by a variety of factors, including product quality [47], service [9], the price [27], product ranking [48], and the degree of knowledge of product information [49,50]. In addition, enterprise image [51], brand perception [28], and online reviews [45] are also important factors affecting consumers' product quality perception. Han et al. [51] found that enterprise ethical attributes have a significant positive impact on the formation of consumers' perceived status. Li et al. [28] explored retailers' optimal own product quality positioning decisions; it was found that retailers would introduce low-quality own products when branded product quality is high. Meng et al. [29] explored the impact of quality perception level on the pricing decisions and profit of supply chain members by establishing an online shopping supply chain that takes into account consumers' quality perception level. Existing research has mostly focused on static environments, with insufficient research on how differences in quality perception under dynamic multi-cycle conditions regulate consumer channel switching behavior.

Strategic consumers dynamically adjust their purchasing channels based on price and perceived product quality. Online retailers have increasingly adopted price discrimination strategies to reward new consumers and incentivize channel-switching behavior in product purchases [24]. Furthermore, consumer segments can be categorized as high-value and low-value groups based on their purchasing patterns [52,53]. Existing research classifies discount strategies into proportional preference and monetary preference [14,54,55]. Coupon distribution can effectively attract new consumers [15], bring stronger consumer engagement [56], increase consumer purchase impulse [57] and promote social empowerment [58], etc. Discount rate and coupon ease of use affect consumer purchase decisions [59]. Zhang et al. [55] empirically demonstrate that proportional discounts are more effective for low-grade products, whereas monetary discounts better suit high-grade products. Zhang et al. [60] established two promotional models, price discount and cash coupon, which are decided by the platform and consider the merchant's advertisement investment strategy, and found that the platform should use the cash coupon strategy when the transaction rate is high, and the price of the products is low. Otherwise, it should use the price discount strategy. By comparing the two scenarios of platform-issued and non-coupon issued, Wang and Yang [61] found that platform-issued coupons are beneficial to the platform and manufacturers' profit but may harm retailers' interests. Chen et al. [62] further reveal that intensified price discounts diminish retailers' incentives to invest in advertising when online and offline channels compete on both price and promotional efforts. Li et al. [63] discussed the issue of issuing coupons in the online channel under omnichannel retailing. They found that retailers should implement a low-price strategy in the pure online channel to attract consumers with high travelling costs and issue coupons with substantial discounts in the BOPS (order online, pick up offline) channel to attract consumers with a high perceived value of product experience. In addition, Li et al. [64] analyzed the optimal promotional strategies of manufacturers and retailers before and after the introduction of private labels by retailers for an omnichannel supply chain providing BOPS services. The implementation of these promotional strategies can stimulate consumers' purchasing behavior to a certain extent and, at the same time, play a role in pulling new products and promotions [15,65]. These studies mostly

assume that consumer valuations are exogenous, ignoring the dynamic impact of differences in quality perceptions on price sensitivity [24]. We combine a two-period model to analyze how consumers' channel switching behavior based on quality perceptions affects product pricing.

The platform operation mode affects the optimal pricing of products to a certain extent [33]. Current research on the choice of platform operation model mainly focuses on the comparison and choice of self-operated model and commission model [30]. The main difference between the two models is that platforms assume different roles in sales [17]. When online and offline channels compete, the commission-based model outperforms the self-operated model. With the intensification of competition between online and offline channels, enterprises will prefer the commission-based model [17]. Consider the case of retail competition between two enterprises that can generate Pareto improvements, one using the self-operated model and the other using the commission-based model [66]. When a supplier does not have a direct sales channel, at least one competing supplier always adopts a commission-based model [67]. In markets with heterogeneous firm competitiveness, the dominant e-retailer should adopt a proprietary model [68]. In addition to channel competition, the commission fee is also an important factor affecting the choice of retailers' operation model [31]. Zhen and Gu [32] considered the showrooming behavior of consumers when purchasing experiential products. They found that when the commission ratio is within a reasonable range, retailers choose the commission-based model if the consumer satisfaction rate of the product is very low, and conversely, they choose the self-operated model. In addition, Wei and Dong [69] derived the conditions under which all four distribution models are in equilibrium and found that the use of pricing power and resale double marginalization plays a key role in the equilibrium outcome. In supply chain games, the order of decision-making significantly affects the equilibrium outcome. Fudenberg et al. [25] pointed out that leaders can obtain higher profits through first-mover advantages, and in e-commerce platform competition, differences in decision-making sequences further interact with operating models. However, existing literature often ignores the joint effect of decision order and quality perception. For example, Zhang Zijian and Liu Wenjing [31] studied pattern selection under channel competition, but did not analyze how differences in consumer quality perception moderate the pricing strategies of leaders and followers. This paper reveals the interactive mechanism between decision-making sequence, quality perception, and operating mode by constructing Stackelberg and Nash equilibrium models.

After combing through the above literature, the literature and factors addressed by our study are summarized in Table 1. It is found that e-commerce platforms, serving as an important channel for online retailing, possess a rich research foundation regarding their operation strategy selection. Pricing decision-making and platform operation mode

**Table 1. The summary of literature and factors addressed by our study.**

| Reference | two-period | consumer preferences | price discrimination | quality perception | platform operations model | channel competition | e-commerce platform |
|---|---|---|---|---|---|---|---|
| Zhao and Wang [3] | | √ | | | √ | √ | √ |
| Wang and Yan [10] | | √ | | | √ | √ | √ |
| Sun et al [14] | | √ | | | | √ | |
| Zhang and Wen [67] | | | | | √ | √ | √ |
| K Matsui [65] | | | | | √ | √ | √ |
| Reimers et al [20] | √ | √ | √ | | | √ | |
| Liu and Zhang [24] | √ | √ | √ | | | √ | |
| Meng et al [47] | | √ | | √ | | √ | √ |
| Ma et al [49] | √ | √ | √ | | √ | √ | √ |
| Zhang et al [63] | √ | | √ | | √ | √ | √ |
| Fudenberg et al [27] | √ | √ | √ | | | √ | |
| Our study | √ | √ | √ | √ | √ | √ | √ |

selection in the context of product quality perception differentiation are currently hot research topics. However, there are still deficiencies in the current research. On the one hand, research on pricing decision-making primarily focuses on product prices and price concessions from the retailers' perspective, with relatively little discussion on how product quality perception influences supply chain pricing decision-making. Especially in the context of channel competition, studies focusing on this issue are even scarcer; on the other hand, most current research on platform operation mode selection concentrates on dual-channel supply chain models that include only a single retailer, with relatively fewer studies involving dual-platform supply chains. It is especially true when the impact of platform operation modes and enterprises' market positions on the order of decision-making is also taken into consideration.

In terms of theory, numerical analysis, and industrial practice, this study demonstrates significant innovations compared with existing research. Theoretically, it breaks through the limitations of existing studies that focus on objective quality improvement or static price discrimination [28,31] while neglecting the dynamic impact of quality perception. It takes the difference in consumers' product quality perception as an endogenous core variable in the two-period dynamic game and explores the impact of quality perception on platform pricing and mode selection in the context of dynamic multi-period scenarios. In contrast, most existing studies are based on single-platform or static models [33] and fail to conduct in-depth exploration of the dynamic interaction between dual platforms. Numerically, different from the single-parameter verification in existing studies, this study conducts multiple sets of simulations (including low/medium/high levels of quality perception difference [24,53] combined with changes in the probability of high-quality perception). It quantifies the dynamic impact of quality perception difference on profits, confirms the feasibility of the "locking in consumers with low prices in the first period and maintaining market share by raising prices in the second period" strategy for leader platforms under sequential decision-making, and verifies the positive profit effect of the expanded perception difference, thus providing accurate support. Industrially, it combines the actual situation of e-commerce, puts forward practical suggestions, and explains phenomena such as "subsidizing first and then raising prices" and "LV's sales boom despite price increases". However, most existing studies only stay at the level of theoretical deduction and lack sufficient integration with the industry.

In view of this, we construct a supply chain model comprising two platforms and two manufacturers, classify consumers based on their different perceived product quality valuations, and analyze the two-period dynamic pricing game between the platforms in the self-operated model and the commission model. Based on this model, we discuss how supply chain members should make pricing decisions to attract consumers to switch purchasing channels in the second period under different game sequences, as well as how platforms should choose the optimal operation strategies. Through multidimensional considerations in the analysis, it provides a theoretical basis and decision-making suggestions for e-commerce platforms to formulate operation strategies, thereby promoting the innovative development of e-commerce platforms and optimizing the e-tailing market.

## 3. Model description

### 3.1. Description of the problem

In this model, we consider a two-period product retail supply chain system consisting of two e-commerce platforms (the platforms) $E_i, i = (A, B)$ and two manufacturers $M_j, j = (A, B)$, where the manufacturer $M_A$ sells product $A$ to consumers through the platform $E_A$ and the manufacturer $M_B$ sells product $B$ to consumers through the platform $E_B$. The products $A$ and the product $B$ are substitutable for each other in the market. Before purchasing a product, the consumer cannot be sure of its actual quality. At the end of the first purchase period, the consumer is informed about the actual quality of the product purchased and chooses the channel for the second purchase period based on the perceived quality of the product. Assuming that consumers' perception of product quality is valued as $\nu$, which is divided into high perception $(H)$ and low perception $(L)$, where the probability of $\nu = H$ is $\lambda$, the probability of $\nu = L$ is $1 - \lambda$, $(0 < \lambda < 1)$, and $\bar{\nu} = \lambda H + (1 - \lambda)L$ denotes the average of product quality valuation. It is assumed that only the part of consumers who obtain a low

perception of product quality in the first period will choose to switch channels to purchase the product [24]; the part of consumers who get a low-quality experience in the first period will decide to switch platforms to buy the product in the second period. The remaining low-quality experience consumers and those who obtain a high-quality experience in the first period will choose to repeat the purchase of the product in the second period.

The platform $E_i$ employs a price discrimination strategy to attract consumers from competing channels (new consumers) by adjusting the product's purchase price in the second period. Since there are two operation modes for platforms to choose from, namely commission-based mode and self-operated mode, and different price discrimination strategies are implemented under different operation modes. Platforms maximize their profit by choosing different operating models. Under the self-operated mode, the platform attracts consumers through price discounts with a price discount factor of $\beta_i^k, i = (A, B), k = (S, D)$, as illustrated in Fig 1. Assume that the price discount factor is exogenous, so that $\beta_A^k = \beta_B^k = \beta^k, k = (S, D)$. For example, Amazon launches LD (Lighting Deal) limited-time special offer seconds. In the commission-based model, the platform gives consumers a discount amount $e_i^k, i = (A, B), k = (S, D)$ to attract consumers, as illustrated in Fig 1. For example, Taobao and Pinduoduo opened tens of billions of subsidy channels.

### 3.2. Symbolic assumptions

The model involves symbols and meanings as shown in Table 2.

### 4. Model construction

According to the above problem description, it can be seen that the platform has the following four strategy combinations to choose from: self-operated model under simultaneous decision-making by both platforms (S-R), commission-based model under simultaneous decision-making by both platforms (S-C), self-operated model under sequential decision-making by both platforms (D-R), and commission-based model under sequential decision-making by both platforms (D-C).

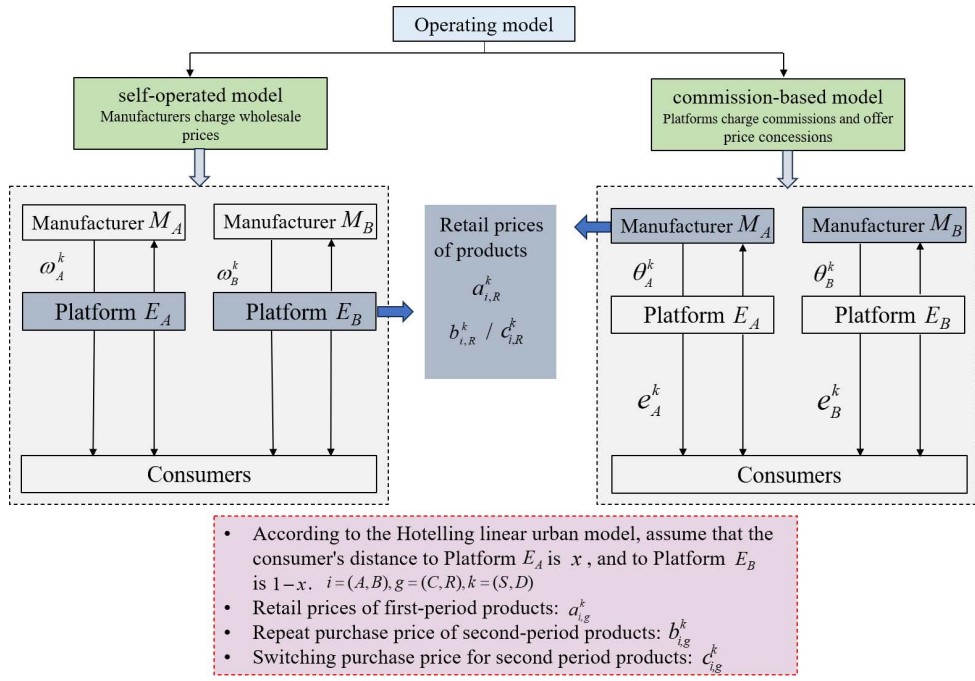

**Fig 1. Schematic diagram of the operating model.**

**Table 2. Parameter values of numerical simulations.**

| Parameter | Definitions |
|---|---|
| $\bar{v}$ | Valuation of consumer expectations of product quality |
| $\lambda$ | Probability that a consumer will receive a high-quality perception ($0 < \lambda < 1$) |
| $g$ | $g = (C, R)$ indicates commission-based and self-operated models, respectively |
| $k$ | $k = (S, D)$ indicates simultaneous and sequential decision-making by the two platforms, respectively |
| $tx$ | Negative utility incurred by consumers purchasing products on the platform |
| $y$ | The point at which the utility of the product $A$ and the product $B$ are indistinguishable to consumers in the first period |
| $x_0$ | Point of no difference in utility between consumers repeating and switching purchases of a product $A$ in the second period |
| $x_1$ | Point of no difference in utility between consumers repeating and switching purchases of a product $B$ in the second period |
| $t$ | unit search cost |
| $\omega_j^k$ | Wholesale prices for Manufacturer $M_j$ decisions |
| $\theta_i^k$ | Commission rate charged by the platform $E_i$ |
| $\beta^k$ | Price discount factor granted by platform $E_i$ of new consumers |
| $e_i^k$ | Amount of price concessions granted by platform $E_i$ of new consumers |
| $d_j$ | Market demand for product $A$ $(B)$ in the first period |
| $D_j$ | Market demand for product $A$ $(B)$ in the second period |
| $a_{i,g}^k$ | Retail prices of first-period products |
| $b_{i,g}^k$ | Repeat purchase price of second-period products |
| $c_{i,g}^k$ | Switching purchase price for second period products |
| $\Pi_{i,g}^k$ | Total Supply Chain Profit |
| $\pi\left(M_{j,g}^k\right)$ | Second-period profit for manufacturers |
| $\Pi\left(M_{j,g}^k\right)$ | Manufacturer's total profit for the second period |
| $\pi\left(E_{i,g}^k\right)$ | Second-period profit of the platform |
| $\Pi\left(E_{i,g}^k\right)$ | Total two-period profit of the platform |

## 4.1. Self-operated model under simultaneous decision making (S-R)

We develop a two-period pricing game under the self-operated mode, where both platforms move simultaneously. The order of the game is as follows: first, the manufacturer $M_j$ decides the wholesale price of the product $\omega_j^S$. Second, the platform $E_i$ decides the retail price of the product $a_{iR}^S$ in the first period. And next, the platform $E_i$ decides the retail price of the product $b_{iR}^S$ for repeat purchases and the retail price of the product $c_{iR}^S$ for switching purchases in the second period, respectively.

By comparing consumer second period utility:

$$L - b_{AR}^S - tx_0 = \bar{v} - c_{BR}^S - t\left(1 - x_0\right) + \beta^k c_{BR}^S \tag{1}$$

$$L - b_{BR}^S - t\left(1 - x_1\right) = \bar{v} - c_{AR}^S - tx_1 + \beta^k c_{AR}^S \tag{2}$$

The utility indistinguishability points for consumers repeating and switching purchases of product $A$ and product $B$ in the second period can be obtained as follows:

$$x_0 = \frac{t - b_{AR}^S - \lambda(H - L) - c_{BR}^S (\beta^S - 1)}{2t}$$

(3)

$$x_1 = \frac{t + b_{BR}^S + \lambda(H - L) + c_{AR}^S (\beta^S - 1)}{2t}$$

(4)

Consumers choose products in the first period with the goal of maximizing the two-period expected utility, the product $A$ first-period market demand is $d_A = y$, and the second-period market demand includes both repeat and switching consumers, i.e., $D_A = \lambda y + (1 - \lambda) x_0 + (1 - \lambda) (x_1 - y)$; similarly, the product $B$ first-period market demand is $d_B = (1 - y)$, and the second-period market demand is $D_B = \lambda(1 - y) + (1 - \lambda) (1 - x_1) + (1 - \lambda) (y - x_0)$. By comparing the two-period utility of consumers can be obtained:

$$\bar{v} - c_{BR}^S - t (1 - x_0) + \beta^S c_{BR}^S + \bar{v} - a_{AR}^S - td_A = (\bar{v} - c_{AR}^S - tx_1 + \beta^S c_{AR}^S) + (\bar{v} - a_{BR}^S - td_B)$$

(5)

Based on the Nash equilibrium, the solution identifies the point at which there is no difference in utility between purchasing the two products in the first period as:

$$y = \frac{t (\lambda - 2) - a_{AR}^S (\lambda - 1) + a_{BR}^S (\lambda - 1)}{2t (\lambda - 2)}$$

(6)

The profit of platform $E_i$ in the second period and the total profit in the two periods are as follows:

$$\pi \left( E_{AR}^S \right) = \left( b_{AR}^S - \omega_A^S \right) \left( \lambda y + (1 - \lambda) x_0 \right) + (1 - \lambda) (x_1 - y) \left( c_{AR}^S - \omega_A^S - \beta^S c_{AR}^S \right)$$

(7)

$$\pi \left( E_{BR}^S \right) = \left( b_{BR}^S - \omega_B^S \right) \left( \lambda (1 - y) + (1 - \lambda) (1 - x_1) \right) + (1 - \lambda) (y - x_0) \left( c_{BR}^S - \omega_B^S - \beta^S c_{BR}^S \right)$$

(8)

$$\Pi \left( E_{AR}^S \right) = \left( a_{AR}^S - \omega_A^S \right) d_A + \left( b_{AR}^S - \omega_A^S \right) \left( \lambda y + (1 - \lambda) x_0 \right) + (1 - \lambda) (x_1 - y) \left( c_{AR}^S - \omega_A^S - \beta^S c_{AR}^S \right)$$

(9)

$$\Pi \left( E_{BR}^S \right) = d_B(a_{BR}^S - \omega_B^S) + (b_{BR}^S - \omega_B^S) \left( \lambda (1 - y) + (1 - \lambda) (1 - x_1) \right) + (1 - \lambda)(y - x_0)(c_{BR}^S - \omega_B^S - \beta^S c_{BR}^S)$$

(10)

The profit of manufacturer $M_j$ in the second period and the total profit in the two periods are as follows:

$$\pi \left( M_{AR}^S \right) = \omega_A^S D_A$$

(11)

$$\pi \left( M_{BR}^S \right) = w_B^S D_B$$

(12)

$$\Pi\left(M_{AR}^{S}\right) = \omega_{A}^{S}d_{A} + \omega_{A}^{S}D_{A} \tag{13}$$

$$\Pi\left(M_{BR}^{S}\right) = \omega_{B}^{S}d_{B} + \omega_{B}^{S}D_{B} \tag{14}$$

The Nash equilibrium solutions are found based on backward induction:

$$\omega_{A}^{S} = \frac{t\left(34 - \lambda\left(19 + 8\lambda\right)\right)}{\left(\lambda - 1\right)\left(2\lambda - 13\right)} \tag{15}$$

$$\omega_{B}^{S} = \frac{t\left(34 - \lambda\left(19 + 8\lambda\right)\right)}{\left(\lambda - 1\right)\left(2\lambda - 13\right)} \tag{16}$$

$$a_{AR}^{S} = \frac{2}{3}\left((H-L)\lambda + \frac{t\left(90 - \lambda\left(67 + 7\lambda\right)\right)}{\left(\lambda - 1\right)\left(2\lambda - 13\right)}\right) \tag{17}$$

$$a_{BR}^{S} = \frac{2}{3}\left((H-L)\lambda + \frac{t\left(90 - \lambda\left(67 + 7\lambda\right)\right)}{\left(\lambda - 1\right)\left(2\lambda - 13\right)}\right) \tag{18}$$

$$b_{AR}^{S} = \frac{1}{3}\left(\frac{t\left(128 - \lambda\left(61 + 24\lambda\right)\right)}{\left(\lambda - 1\right)\left(2\lambda - 13\right)} - (H-L)\lambda\right) \tag{19}$$

$$b_{BR}^{S} = \frac{1}{3}\left(\frac{t\left(128 - \lambda\left(61 + 24\lambda\right)\right)}{\left(\lambda - 1\right)\left(2\lambda - 13\right)} - (H-L)\lambda\right) \tag{20}$$

$$c_{AR}^{S} = \frac{\frac{t\left(\lambda\left(59+24\lambda\right)-115\right)}{\left(\lambda-1\right)\left(2\lambda-13\right)} - (H-L)\lambda}{3\left(\beta^{S} - 1\right)} \tag{21}$$

$$c_{BR}^{S} = \frac{\frac{t\left(\lambda\left(59+24\lambda\right)-115\right)}{\left(\lambda-1\right)\left(2\lambda-13\right)} - (H-L)\lambda}{3\left(\beta^{S} - 1\right)} \tag{22}$$

It can be shown that $b_{AR}^{S} = b_{BR}^{S}$, $c_{AR}^{S} = c_{BR}^{S}$, $a_{AR}^{S} = a_{BR}^{S}$, $\omega_{A}^{S} = \omega_{B}^{S}$, the optimal solutions of the platform $E_{A}, E_{B}$ are all the same in the S-R case.

## 4.2. Commission-based model under simultaneous decision-making (S-C)

We construct a pricing game model under the two-period commission-based mode, in which two platforms make decisions simultaneously. The order of the game is as follows: first, the platform $E_i$ decides on the commission rate $\theta_i^S$. Second, the manufacturer $M_j$ decides on the retail price of the product $a_{iC}^S$ in the first period. Next, the platform $E_i$ decides on the price discount of the new consumers $e_i^S$. And lastly, the manufacturer $M_j$ decides on the retail price of the product $b_{iC}^S$ in the second period for repeat purchases and $c_{iC}^S$ in the second period for switching purchases, respectively.

Unlike the self-operated model, in the commission-based model, the manufacturer controls product pricing, while the platform decides the commission rate. Similar to 4.1, the consumer utility function, manufacturer and platform profit functions, and the game equilibrium solutions are provided in Appendix S2 in S1 Appendix.

## 4.3. Self-operated model under sequential decision-making (D-R)

We establish a two-period pricing game under the self-operated mode, where platforms decide sequentially. The order of the game is as follows: first, the manufacturer $M_j$ decides the wholesale price of the product $\omega_j^D$ respectively. Second, the platform $E_A$ decides the retail price of the product $a_{AR}^D$ in the first period, and then the platform $E_B$ decides the retail price of the product $a_{BR}^D$ in the first period. Next, the platform $E_A$ decides the retail price of the product $b_{AR}^D$ in the second period for repeat purchases and the retail price of the switching product $c_{AR}^D$. And then the platform $E_B$ decides the retail price of the product $b_{BR}^D$ in the second period for repeat purchases and the retail price of the switching product $c_{BR}^D$.

Based on Equations (1) and (2), the utility indifference points for consumers choosing between repeat purchases and switching purchases in the second cycle are as follows:

$$x_0 = \frac{t - \lambda(H-L) - b_{AR}^D - c_{BR}^D \left(\beta^D - 1\right)}{2t} \tag{23}$$

$$x_1 = \frac{t + \lambda(H-L) + c_{AR}^D \left(\beta^D - 1\right) + b_{BR}^D}{2t} \tag{24}$$

From Equation (5), it can be obtained that there is no point of difference in the utility of the two products purchased by consumers in the first period:

$$y = \frac{\lambda\left(H-L\right)\left(\lambda-1\right) + t\left(4\lambda-9\right) - 4a_{AR}^D\left(\lambda-1\right) + 4a_{BR}^D\left(\lambda-1\right)}{t\left(8\lambda-19\right)} \tag{25}$$

The profit of self-operated platform $E_i$ in the second period and the total profit in the two periods are as follows:

$$\pi\left(E_{AR}^D\right) = \left(b_{AR}^D - \omega_A^D\right)\left(\lambda y + (1-\lambda)x_0\right) + (1-\lambda)\left(x_1 - y\right)\left(c_{AR}^D - \omega_A^D - \beta^D c_{AR}^D\right) \tag{26}$$

$$\pi\left(E_{BR}^D\right) = \left(b_{BR}^D - \omega_B^D\right)\left(\lambda\left(1-y\right) + (1-\lambda)\left(1-x_1\right)\right) + (1-\lambda)\left(y - x_0\right)\left(c_{BR}^D - \omega_B^D - \beta^D c_{BR}^D\right) \tag{27}$$

$$\Pi\left(E_{AR}^D\right) = \left(a_{AR}^D - \omega_A^D\right)y + \pi\left(E_{AR}^D\right) \tag{28}$$

$$\Pi\left(E_{BR}^D\right) = \left(a_{BR}^D - \omega_B^D\right)(1-y) + \pi\left(E_{BR}^D\right) \tag{29}$$

The profit of manufacturer $M_j$ in the second period and the total profit in the two periods are as follows:

$$\pi\left(M_{AR}^D\right) = \omega_A^D\left(\lambda y + (1-\lambda)\, x_0 + (1-\lambda)\,(x_1 - y)\right) \tag{30}$$

$$\pi\left(M_{BR}^D\right) = \omega_B^D\left(\lambda\,(1-y) + (1-\lambda)\,(1-x_1) + (1-\lambda)\,(y - x_0)\right) \tag{31}$$

$$\Pi\left(M_{AR}^D\right) = \omega_A^D y + \pi\left(M_{AR}^D\right) \tag{32}$$

$$\Pi\left(M_{BR}^D\right) = \omega_B^D\,(1-y) + \pi\left(M_{BR}^D\right) \tag{33}$$

The equilibrium solutions of this Stackelberg model game obtained using backward induction are as follows:

$$\omega_A^D = \frac{\left(\lambda\,(H-L)\,(1-\lambda)\,(3+2\lambda)\right) + t\,(899 + \lambda\,(4\,(\lambda-41)\,\lambda - 545))}{3\,(\lambda-1)\,(4\lambda\,(5+\lambda) - 95)} \tag{34}$$

$$\omega_B^D = \frac{1021t + \lambda\,(H-L)\,(\lambda-1)\,(3+2\lambda) - t\lambda\,(607 + 4\,(55+\lambda))}{3\,(\lambda-1)\,(4\lambda\,(5+\lambda) - 95)} \tag{35}$$

$$a_{AR}^D = \frac{\lambda(H-L)(\lambda-1)(72705 + 2(\lambda(4\lambda(628+83\lambda)-4527)-29369))}{192(\lambda-1)(\lambda(3+\lambda)-5)(4\lambda(5+\lambda)-95)} + \frac{t(\lambda(779939 + 2\lambda(\lambda(972\lambda^2 + 8\lambda^3 - 918\lambda - 60069) - 57735))) - 561665}{192(\lambda-1)(\lambda(3+\lambda)-5)(4\lambda(5+\lambda)-95)} \tag{36}$$

$$a_{BR}^D = \frac{11147t - \frac{11558t}{\lambda-1} + 71\lambda\,(59H - 59L + 2t) + \frac{142\,(17t + \lambda\,(L-H+4t))}{\lambda(3+\lambda)-5} - \frac{24\,(497\lambda\,(L-H) + 4t\,(18446 + 4589\lambda))}{4\lambda(5+\lambda)-95}}{6816} \tag{37}$$

$$b_{AR}^D = \frac{\left(\lambda\,(H\text{-}L)\,(1\text{-}\lambda)\,(11145 + \lambda\,(4\lambda\,(3\lambda\,(67+8\lambda)\,\text{-}292)\,\text{-}9379)))\right)}{48\,(\lambda-1)\,(\lambda\,(3+\lambda)-5)\,(4\lambda\,(5+\lambda)-95)} + \frac{t\,(\lambda\,(109188 + \lambda\,(3293 + 2\lambda\,(2\lambda\,(18\lambda-653)-9485)))) - 98215)}{48\,(\lambda-1)\,(\lambda\,(3+\lambda)-5)\,(4\lambda\,(5+\lambda)-95)} \tag{38}$$

$$b_{BR}^D = \frac{t\left(195450 + \lambda\left(\lambda\left(2\lambda\left(21133 + 3898\lambda + 68\lambda^2\right) - 13833\right) - 221737\right)\right)}{96\,(1-\lambda)\,(\lambda\,(3+\lambda)-5)\,(4\lambda\,(5+\lambda)-95)} + \frac{t\left(195450 + \lambda\left(\lambda\left(2\lambda\left(21133 + 3898\lambda + 68\lambda^2\right) - 13833\right) - 221737\right)\right)}{96\,(1-\lambda)\,(\lambda\,(3+\lambda)-5)\,(4\lambda\,(5+\lambda)-95)} \tag{39}$$

$$c_{AR}^D = \frac{\left(\lambda\,(H-L)\,(1-\lambda)\,(10890 + \lambda\,(4\lambda\,(3\lambda\,(61+8\lambda)-341)-8753))\right)}{48\,(\beta^D-1)\,(\lambda-1)\,(\lambda\,(3+\lambda)-5)\,(4\lambda\,(5+\lambda)-95)} + \frac{t\,(90970 + \lambda\,(\lambda\,(2\lambda\,(9971 + 2\,(683-18\lambda)\,\lambda)-719)-108459))}{48\,(\beta^D-1)\,(\lambda-1)\,(\lambda\,(3+\lambda)-5)\,(4\lambda\,(5+\lambda)-95)} \tag{40}$$

$$c^D_{BR} = \frac{(\lambda(H-L)(1-\lambda)(11925+\lambda(4\lambda(\lambda(209+24\lambda)-292)-9859)))}{96(\beta^D-1)(\lambda-1)(\lambda(3+\lambda)-5)(4\lambda(5+\lambda)-95)} + \frac{t(177125+\lambda(\lambda(2\lambda(19299+3670\lambda+68\lambda^2)-14603)-198312))}{96(\beta^D-1)(\lambda-1)(\lambda(3+\lambda)-5)(4\lambda(5+\lambda)-95)}$$

(41)

#### 4.4. Commission-based model under sequential decision-making (D-C)

We construct a two-period commission-based pricing game model, in which the two platforms make decisions successively. The order of the game is as follows: first, the platform $E_A$ decides on the commission rate $\theta^D_A$, and then the platform $E_B$ decides on the commission rate $\theta^D_B$. Second, the manufacturer $M_j$ decides on the retail price of the product in the first period $a^D_{iC}$. Next, the platform $E_A$ decides on the amount of price preference to be given to the new consumer $e^D_A$, and then the platform $E_B$ decides on the amount of price preference to be given to the new consumer $e^D_B$. Finally, the manufacturer $M_j$ decides on the retail price of the product for the second period of repeat purchases $b^D_{iC}$ and for the switching purchases $c^D_{iC}$, respectively.

The leader platform $E_A$ makes decisions first, and the follower platform $E_B$ makes decisions afterward. Similar to 4.3, the consumer utility function, manufacturer and platform profit functions, and the game equilibrium solution are provided in Appendix S4 in S1 Appendix.

### 5. Equilibrium analysis

**Theorem 1.** When platforms choose a commission-based model, manufacturers' first-period product prices, second-period repeat product prices, and switching purchase product prices are all positively correlated with the probability of consumers obtaining a high-quality perception, regardless of whether competing platforms make simultaneous or sequential decisions, $\frac{\partial a^k_{iC}}{\partial \lambda} > 0$, $\frac{\partial b^k_{iC}}{\partial \lambda} > 0$, $\frac{\partial c^k_{iC}}{\partial \lambda} > 0$.

The relevant proofs are presented in Appendix S5 in S1 Appendix. According to Theorem 1, as the value of $\lambda$ increases, the manufacturer's prices for first-period products, repeat-purchase products in the second period, and switch-purchase products all exhibit a positive correlation with the probability of consumers obtaining a high-quality perception. When consumers' likelihood of obtaining high-quality perception rises, they are more inclined to repurchase products from the same brand, thereby becoming loyal customers. For instance, Apple's iPhone users are more likely to repurchase a new product upon its release, having experienced the high performance and quality user experience, even if the price is relatively high. As the probability of consumers obtaining high-quality perception increases, their sensitivity to price diminishes, and they focus more on the quality and performance of the product. Consequently, manufacturers can raise repeat-purchase prices to extract greater profits. With increasing consumer demand for high-quality products, a manufacturer's higher prices not only offset the additional costs associated with higher product quality but also establish a high-quality product positioning in the competitive marketplace, attracting and retaining loyal customers. In the context of a commission-based model, when the commission rate remains constant, as the value of $\lambda$ increases, the manufacturer can maximize profit by appropriately raising the retail price of the product. For example, in February 2022, following the Chinese New Year, the luxury brand Louis Vuitton (LV) increased the prices of its entire product line, and surprisingly, its sales increased rather than decreased [70].

Management implications: When a manufacturing enterprise collaborates with a commission-based platform, it can bolster technological innovation and motivate the platform to enhance consumers' perception of product quality by accelerating digital transformation, reinforcing quality control, and establishing a feedback mechanism for quality signaling. Furthermore, the manufacturing enterprise has the opportunity to moderately elevate product prices throughout the full-period purchasing process, thereby achieving higher profit.

**Theorem 2.** When a platform chooses the self-operated model, the platform's first-period product prices, second-period repeat product prices, and switching purchase product prices are positively correlated with the probability of consumers

obtaining a high-quality perception, regardless of whether decisions are made at the same time or sequentially among competing platforms, $\frac{\partial a_{iR}^k}{\partial \lambda} > 0$, $\frac{\partial b_{iR}^k}{\partial \lambda} > 0$, $\frac{\partial c_{iR}^k}{\partial \lambda} > 0$.

Complete proofs are available in Appendix S6 in S1 Appendix. From Theorem 2, When the value of $\lambda$ gradually increases, the prices of the platform's first-period products, the product prices of repeating purchases, and the product prices of switching purchases in the second period all increase. As $\lambda$ increases, it indicates that the more consumers get high-quality perceived value, the higher consumers' satisfaction with the product, and the stronger their purchase intention. Increasing the retail price of a product can attract price-insensitive, quality-conscious consumers while maintaining existing customer loyalty. For example, Amazon implements a dynamic pricing strategy, using big data and AI technology to monitor market demand and competitors' prices in real-time, and is able to quickly adjust product prices when the probability of consumers obtaining high-quality perception increases, increasing profit by raising product prices. In the self-managed model, when the wholesale price of the product is certain, as $\lambda$ increases, the platform can increase profit by increasing product prices.

Management implications: When a manufacturing enterprise collaborates with a self-managed platform, it can enhance product quality by optimizing raw material selection, refining manufacturing processes, investing in research and development (R&D), and strengthening quality control measures. These initiatives, in turn, enhance consumers' perceived product quality. Subsequently, the platform may consider raising prices to realize higher profit.

From Theorems 1 and 2, it is clear that both manufacturing enterprises and platforms can earn higher profit by increasing the price of their products when the probability of consumers obtaining high-quality perception is high, both in the commission-based (manufacturer-decision price) model and in the captive (platform-decision price) model.

**Theorem 3.** In the commission-based model, the commission rate is higher when platforms make sequential decisions than when platforms make simultaneous decisions, $\theta_A^D > \theta_i^S$, $\theta_B^D > \theta_i^S$. In this case, the commission rate of the leader platform is higher than the commission rate of the follower platform in the case of sequential decision-making, $\theta_A^D > \theta_B^D$.

The relevant proofs are presented in Appendix S7 in S1 Appendix. According to Theorem 3, it can be observed that the commission rate is higher when platforms make decisions successively compared to when they make decisions simultaneously. Furthermore, within a successive decision-making scenario, the leader platform charges a higher commission rate than the follower platform. When platforms decide simultaneously, the market competition remains in a relatively stable state, with no single platform dominating the market advantage. Consequently, the commission rate among competitors remains relatively stable and thus lower. In contrast, when platforms decide sequentially, the leader platform enjoys a market advantage, possessing bargaining power and influence. This strategy allows the platform to expand its consumer base and levy higher commission rates on manufacturers. On the other hand, follower platforms must adjust their strategies based on the commission rate set by the leader platform in order to maintain their relative competitiveness. Typically, their commission rate is lower than that of the leader platform. For instance, among commission-based platforms, Tmall, as the market leader, charges a commission rate of 5% for jewellery and accessories products, which is slightly higher than that of other follower platforms, such as Jitterbug.

Management implications: platforms that are market leaders can leverage their market influence to set higher commission rates, thereby increasing their profitability.

**Theorem 4.** In the commission-based model, the amount of discount of new consumers in the second period when the platform makes successive decisions is higher than the amount of discount of new consumers in the second period when the platform makes simultaneous decisions, $e_A^D > e_i^S$, $e_B^D > e_i^S$, where the amount of discount of the leader's platform when making successive decisions is greater than the amount of discount of the follower's platform, $e_A^D > e_B^D$.

The relevant proofs are shown in Appendix S8 in S1 Appendix. From Theorem 4, it can be seen that in the second period of consumer purchases, the amount of price preference given to new consumers by platforms in sequential decision-making is higher than that given in simultaneous decision-making. The leader platform offers a higher discount than the follower platform when making sequential decisions. Under the commission-based model, the leader platform

should leverage its market advantage to set higher discounts to maintain its market share. In contrast, the follower platforms can differentiate themselves by offering moderate discounts to attract specific groups of consumers.

Management implications: The higher the commission a platform charges a manufacturer, the greater its ability to offer higher price concessions to attract price-sensitive consumers to switch and purchase in the second period. At the same time, signaling its capital strength to the market enhances the partnership between the manufacturer and the platform, creating a virtuous period that promotes better market development and increased consumer satisfaction.

**Theorem 5.** In the self-managed model, when platforms make decisions sequentially, the following relationship between the retail prices of products of the leader platform and the follower platform can be obtained:

$$\begin{cases} a_{AR}^D > a_{BR}^D, b_{AR}^D < b_{BR}^D, c_{AR}^D > c_{BR}^D, 0 < \frac{t}{H-L} < l2 \\ a_{AR}^D > a_{BR}^D, b_{AR}^D > b_{BR}^D, c_{AR}^D > c_{BR}^D, l2 < \frac{t}{H-L} < l1 \\ a_{AR}^D < a_{BR}^D, b_{AR}^D > b_{BR}^D, c_{AR}^D > c_{BR}^D, l1 < \frac{t}{H-L} < l3 \\ a_{AR}^D < a_{BR}^D, b_{AR}^D > b_{BR}^D, c_{AR}^D < c_{BR}^D, \frac{t}{H-L} > l3 > 0 \end{cases}$$

Among them:

$$l1 = \frac{\lambda\,(1-\lambda)\,(17955 + 4\lambda\,(\lambda\,(24\lambda\,(13+2\lambda)-721)-3559))}{\lambda\,(55743 + 2\lambda\,(2\lambda\,(2\lambda\,(421+72\lambda)-1727)-15979))-14915}$$

$$l2 = \frac{\lambda\,(\lambda-1)\,(10140 + \lambda\,(4\lambda\,(\lambda\,(227+24\lambda)-171)-9557))}{\lambda\,(\lambda\,(2\lambda\,(2163+2\lambda\,(643+70\lambda))-7247)-3361)-980}$$

$$l3 = \frac{\lambda\,(\lambda-1)\,(9855 + \lambda(4\lambda\,(\lambda\,(157+24\lambda)-390)-7647))}{\lambda(18606 + \lambda(2\lambda\,(14\lambda\,(67+10\lambda)-643)-13165))-4815}$$

The relevant proofs are shown in Appendix S9 in S1 Appendix, and the value of $t/(H-L)$ is divided into four intervals, as illustrated in Fig 2. In interval I, the leader platform's first-period product prices and second-period switching purchase product prices are higher than those of the follower platform, and the leader platform's second-period repeat purchase

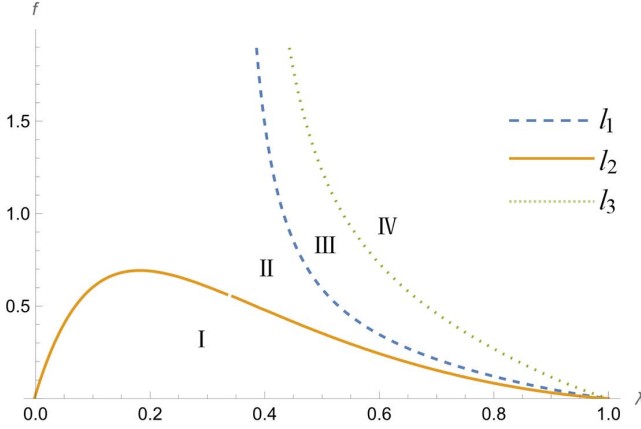

**Fig 2. Distribution of f intervals in Self-operated model.**

product prices are lower than those of the follower platform. In interval II, the leader platform's first-period product prices, second-period repeat purchase product prices, and second-period switch purchase product prices are higher than the follower platform. This is because the leader platforms have greater market share and stronger brand presence and have the ability to set higher prices. In addition, consumers will be willing to pay higher prices due to branded product recognition and enterprise reputation. In interval III, the leader platform has lower first-period product prices than the follower platform and higher second-period switching purchase product prices and second-period repeat purchase product prices than the follower platform. In interval IV, the leader platform's first-period product prices and second-period switch purchase product prices are lower than the follower platform, and the leader platform's second-period repeat purchase product prices are higher than the follower platform.

By comparing interval I and interval II, it can be found that there is a significant difference in the leader platform's second-period repeat purchase product prices under different market conditions. In interval I, the leader platform is more conservative in prices, and its second-period repeat purchase product prices are lower than that of the follower platform, thereby enhancing customer stickiness. In interval II, the leader platform's second-period repeat purchase product prices are higher than those of the follower platform, and consumers are less likely to switch purchasing channels because of the price difference when search costs are high and perceived differences are small.

In intervals III and IV, consumers have a higher probability of obtaining high-quality perception. In these two intervals, the follower platform's first-period product prices are higher than those of the leader platform, and the leader platform adopts a low-price strategy to attract more consumers. Due to the lower prices in the first period, the leader platform has more room for price increases in the second period. Specifically, in interval III, the leader platform's prices for repeating and switching purchases are higher than the follower platform's in the second period, thereby increasing the platform's profit. However, in interval IV, the leader platform's second-period switching purchase prices are lower than that of the follower platform. When the probability of consumers obtaining high-quality perception is sufficiently high, the leader must significantly reduce the switching purchase price to attract new consumers while demonstrating its higher competitiveness.

Management implications: in the context of platform enterprise competition, when a significant number of consumers have obtained high-quality perception, the leader platform can adopt a lower pricing strategy in the first period to attract consumers. Once a certain degree of customer stickiness or loyalty is formed, in the second period, the prices of repeat purchases can be set higher than that of the follower platform, and consumers may continue to choose the original platform.

**Theorem 6.** In the commission-based model, the relationship between the amount of price preference given by the platform to new consumers and $\lambda$ can be obtained when the platform makes successive decisions as follows, where $f_2 > f_1$. The values of $f_1$ and $f_2$ are illustrated in the Appendix S10 in S1 Appendix.

$$\begin{cases} \frac{\partial e_A^D}{\partial \lambda} > 0, \frac{\partial e_B^D}{\partial \lambda} > 0, \frac{t}{H-L} > f2 > 0 \\ \frac{\partial e_A^D}{\partial \lambda} > 0, \frac{\partial e_B^D}{\partial \lambda} < 0, f1 < \frac{t}{H-L} < f2 \\ \frac{\partial e_A^D}{\partial \lambda} < 0, \frac{\partial e_B^D}{\partial \lambda} < 0, 0 < \frac{t}{H-L} < f1 \end{cases}$$

The relevant proofs are shown in Appendix S10 in S1 Appendix, and the value of $t/(H-L)$ is divided into three intervals, as illustrated in Fig 3, when two commission-based platforms make decisions successively. In interval I, the amount of preference given to new consumers in the second period by both the leader platform and the follower platform increases as the probability of consumers obtaining high-quality perception increases. In interval II, the amount of preference given to new consumers in the second period by the leader platform increases as the probability of consumers. The amount of preference given to new consumers by the leader platform in the second period increases as the probability of consumers obtaining high-quality perception increases, while the amount of preference given to new consumers by the follower platform in the second period decreases as the probability of consumers obtaining high-quality perception increases. In

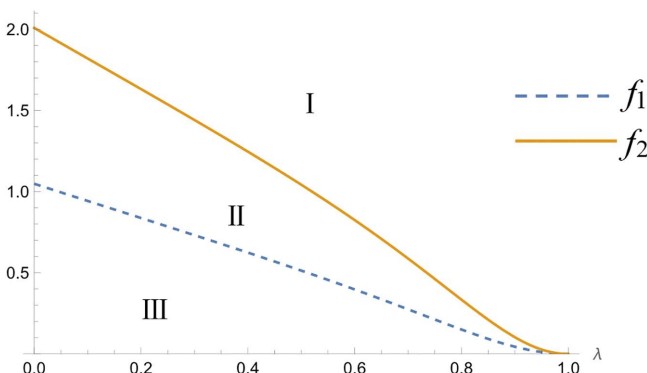

**Fig 3. Distribution of f intervals in the commission-based model.**

interval III, the amount of preference given to new consumers by the leader platform in the second period decreases as the probability of consumers obtaining high-quality perception decreases for both leader platforms and follower platforms.

From Theorem 6, regardless of whether the two platforms make decisions at the same time or sequentially when the difference in consumers' perception of product quality is small, the number of benefits given by the platforms to new consumers in the second period is positively related to the probability of consumers obtaining high product quality perception. When the dispersion in consumers' quality perceptions is large, the second-period benefits offered to new consumers vary inversely with the likelihood that consumers perceive the product as high quality. When the difference in consumers' perception of product quality is small, platforms attract consumers to switch purchases by offering discounts in the second period. Conversely, when the difference is large, platforms do not need to offer higher discounts because consumers base their choices on purchase quality.

Management implications: as consumer expectations of product quality continue to rise, enterprises can adopt a differentiation strategy to compete in the marketplace. Either they can attract price-sensitive consumers by increasing the number of discounts, or they can reduce the number of discounts appropriately and focus on maintaining brand value and increasing profit.

## 6. Numerical simulation

In this section, in order to more graphically analyze the impact of consumer product quality perception differences and channel competition on product prices as well as platform operation mode selection, we will use numerical simulation to validate the research conclusions and to explore the impact of individual parameter changes on supply chain equilibrium decision-making and optimal profit.

### 6.1. Differences in consumer quality perception $H-L$ impact on platform profit

We analyze the impact of consumers' quality perception differences on platform profit when platforms make simultaneous and sequential decisions. Assign values to the unit search cost of online shopping ($t$) and the difference in consumers' quality perception of products ($\Delta s = H-L$), set the relevant parameters $t = 1$, and make $\Delta s = 0.1$, $\Delta s = 5$, and $\Delta s = 10$, respectively, to represent the low, medium, and high levels of the difference in product quality perception [24,53,71]. The probability of consumers obtaining high-quality perception $\lambda \in (0, 1)$, draw the profit curves of commission-based platforms and self-operated platforms in different cases. We obtain Fig 4, Fig 5(a) and Fig 5(b).

As illustrated in Fig 4, the above figure shows the impact of consumer quality perception differences on platform profit when two platforms make decisions at the same time. The dashed line represents the platform profit under the

$\Pi(E_{i,g}^S)$

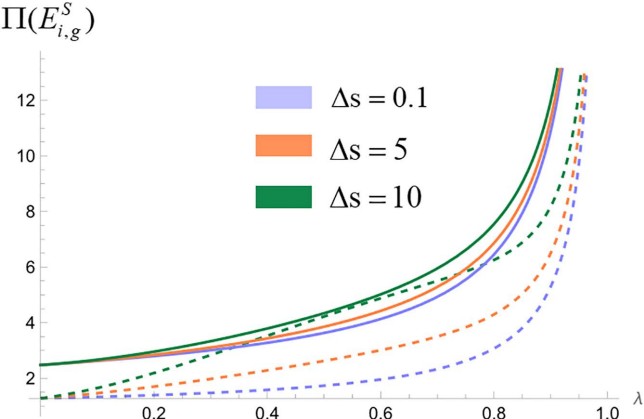

**Fig 4. Impact of differences in consumer quality perception on platform profit when platforms make simultaneous decisions.** ("—"indicates the platform profit in the com-mission-based model, "---" indicates the platform profit in the self-managed model).

$\Pi(E_{Ag}^D)$ $\Pi(E_{Bg}^D)$

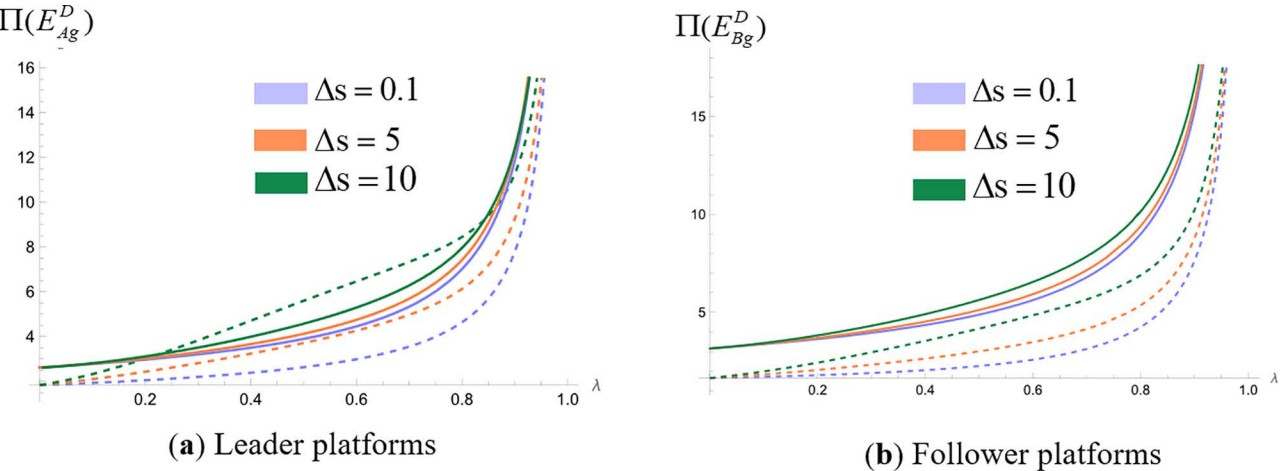

**(a)** Leader platforms **(b)** Follower platforms

**Fig 5. Impact of differences in consumer quality perception on platform profit when platforms make sequential decisions.** ("—"indicates the platform profit in the com-mission-based model, "---" indicates the platform profit in the self-managed model).

self-operated model, while the solid line represents the platform profit under the commission-based model. The three different colors correspond to low, medium, and high levels of perceived quality differences. Due to the symmetry of the best response functions of the two platforms in the 'simultaneous decision' strategy combination, the Nash equilibrium solution is unique and identical, so only six distinguishable profit curves are actually plotted in the figure. Fig 4 shows the profits of self-operated platforms and commission-based platforms under three different quality perception values. From the figure, it can be seen that platform profit increases with the increase of consumer quality perception difference regardless of whether the platform chooses a self-managed model or a commission model.

As illustrated in Fig 5, the above figure shows the impact of consumer quality perception differences on platform profit when two platforms make decisions successively. Fig 5(a) represents the profits of self-operated leader platforms and commission-based leader platforms under three different values of perceived quality differences. Fig 5(b) represents the profits of self-operated follower platforms and commission-based follower platforms under three values of

perceived quality difference. As can be seen from the figure, leader platform profit and follower platform profit increase with the increase of consumer quality perception difference regardless of the platform's choice of self-managed model or commission-based model.

In summary, regardless of whether platforms make simultaneous or sequential decisions and whether they choose self-managed or commission-based models, increased differences in consumers' perception of product quality will increase platform profit. The increase in consumers' perception of product quality difference means that the gap between consumers' expectation of quality and their actual experience becomes bigger, which directly affects consumers' satisfaction and loyalty. When consumers' perception of product quality improves, they will have a higher tendency to repeat purchasing behavior, attract new customers through word-of-mouth recommendations, and help build a strong brand image, and the increased brand value can bring higher premiums and market share for the platform. In addition, as the difference in consumers' perception of quality increases, platforms can more accurately segment the market, adopt differentiated pricing strategies, provide customized products and services for different consumer groups, meet specific needs, increase conversion rates and customer life period value, and thus achieve higher economic benefits.

## 6.2. Differences in consumer perception of quality $H - L$ impact on manufacturers' profit

We analyze the impact of consumers' quality perception difference on manufacturers' profit when platforms make simultaneous and sequential decisions. Assign values to the unit search cost of online shopping ($t$) and the consumer's quality perception difference of the product ($\Delta s = H - L$) respectively, set the relevant parameters $t$ = 1, and make $\Delta s$ = 0.1, $\Delta s$ = 5, and $\Delta s$ = 10, respectively, to represent the low, medium, and high levels of the product quality perception difference. Consumers get the probability of high-quality perception $\lambda \in (0, 1)$, and draw the manufacturer's profit curve in different cases. We obtain Figs 6 and 7 (a), 7 (b), 7 (c) and 7 (d).

When the platforms make simultaneous decisions, the profit function of the manufacturer in the captive model is:

$$\Pi\left(M_{jR}^{S}\right) = \frac{t\left(34 - \lambda\left(19 + 8\lambda\right)\right)}{\left(\lambda - 1\right)\left(2\lambda - 13\right)}$$

(42)

From the above equation, it can be seen that when two platforms make decisions at the same time, if the platform chooses the self-management type model, the difference in the perceived quality of consumer products will not affect the manufacturer's profit. Fig 6 shows the manufacturer's profit under three values of perceived quality difference when two platforms make decisions at the same time and choose the commission-type model. As can be seen from the figure, the manufacturer's profit increases with the increase of the consumer's perceived quality difference of the product.

As illustrated in Fig 7, the upper figure shows the impact of consumer quality perception difference on the manufacturer's profit when the two platforms make decisions successively. Figs 7(a) and 7(b) represent the manufacturer's profit from cooperating with the self-managed leader platform and the follower platform under the three values of the quality perception difference, respectively, and Figs 7(c) and 7(d) represent the manufacturer's profit of cooperating with the commission type leader platforms and follower platforms in terms of manufacturer profit, respectively. As can be seen from the figures, when the platforms make decisions successively, regardless of the operation mode chosen by the platforms, the profit of manufacturers working with the platforms increases with the increase in the perceived difference in consumer product quality.

In summary, regardless of whether the platforms make simultaneous or sequential decisions and whether they choose a self-managed or commission-based model, increased differences in consumers' perception of product quality will increase manufacturers' profit. Increased perceived quality differences can help manufacturers to more accurately conduct market segmentation and product positioning and implement differentiated pricing strategies to meet different consumer needs. In addition, strategic consumers will weigh the utility of purchasing in the face of quality perception differences.

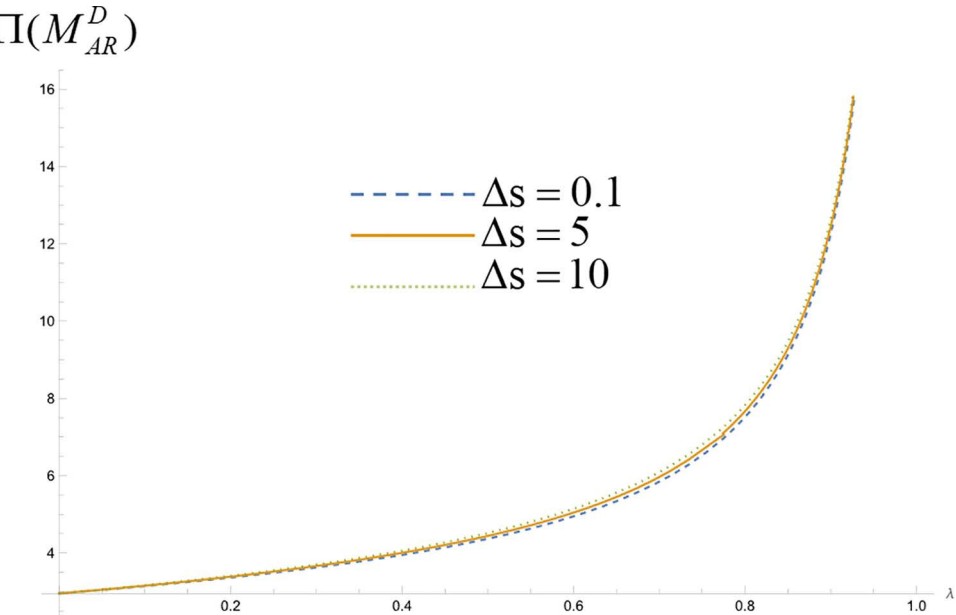

$\Pi(M_{AR}^{D})$

**Fig 6. Impact of differences in consumer quality perception on manufacturer's profitability when platforms make simultaneous decisions.**

Manufacturers can increase production efficiency and reduce costs by disclosing product quality, communicating product value, and continuously innovating and improving products to satisfy consumers' demand for high-quality products, which will motivate consumers to actively purchase high-quality products and, in turn, increase enterprise profit.

### 6.3. Selection of platform operation mode

We analyze which operating model the platform should choose for higher profitability when making simultaneous and sequential decisions. Assign values to the unit search cost of online shopping $(t)$ and the difference in consumers' quality perception of products $(\Delta s = H - L)$ respectively, set the relevant parameter $t = 1$, so that $\Delta s = 0.1$, $\Delta s = 5$, and $\Delta s = 10$, respectively, represent the low, medium and high levels of the difference in the perception of product quality. The probability of consumers obtaining high-quality perception $\lambda \in (0, 1)$. We draw the profit images of commission-based platforms and self-operated platforms in different cases and obtain Figs 8(a), 8(b) and 8(c), 9(a), 9(b) and 9(c), 10(a), 10(b) and 10(c), 11(a), 11(b) and 11(c).

Fig 8(a), 8(b) and 8(c) represent the commission-based platform profit and self-operated platform profit when the two competing platforms make decisions at the same time, and the perceived difference in product quality is small, the perceived difference in product quality is medium, and the perceived difference in product quality is large, respectively. As can be seen from the figure, when competing platforms make decisions simultaneously, regardless of whether consumers' perceived differences in product quality are low, medium, or high, the profit of a platform that adopts the commission-based model is greater than that of a platform that adopts the self-managed model. This is because under the self-managed model, the platform holds the right to set commodity prices, while under the commission-based model, the manufacturer sets product prices. The manufacturer will adjust the retail prices of products to balance market demand, thus maintaining the price level while ensuring product sales volume, so as to maximize its profits.

Figs 9(a), 9 (b) and 9 (c) show the profit of the commission-based leader platform and the profit of the self-owned leader platform when the two leader platforms make decisions successively. The perceived difference in product quality is small, the perceived difference in product quality is medium, and the perceived difference in product quality is large, respectively.

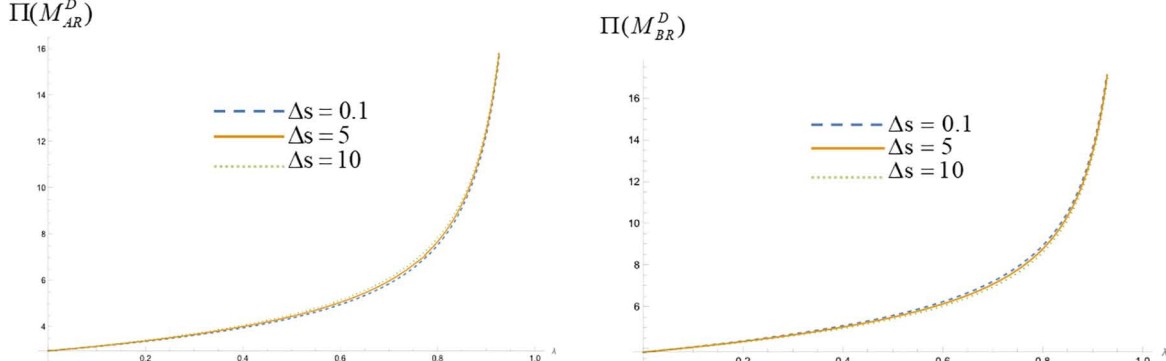

**(a)** Manufacturer profit with captive leader platforms  **(b)** Manufacturer profit with captive follower platforms

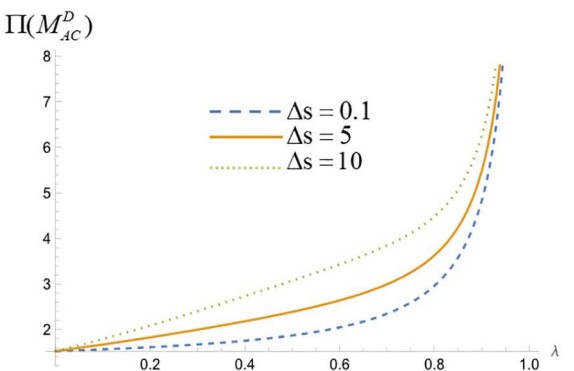

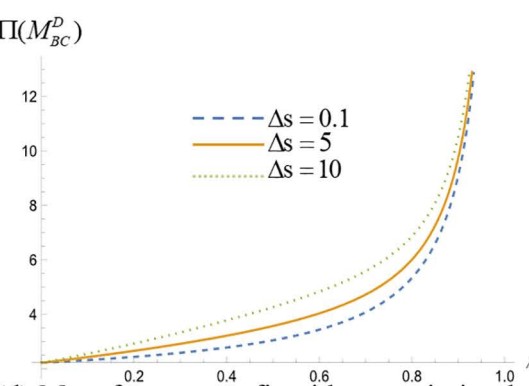

**(c)** Manufacturer profit with commission-based leader platforms

**(d)** Manufacturer profit with commission-based follower platforms

**Fig 7. Impact of differences in consumer quality perception on manufacturers' profitability at platform sequential decision making.**

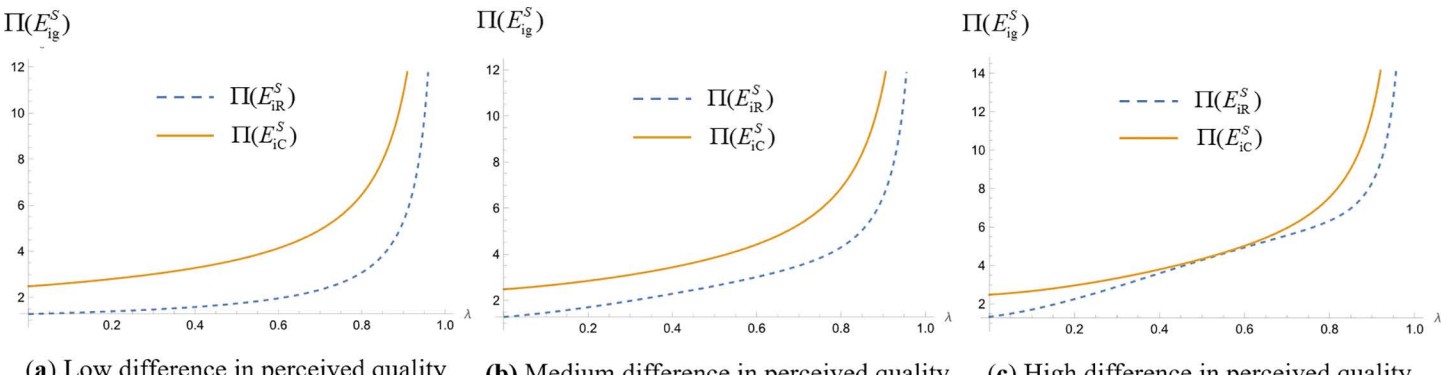

**(a)** Low difference in perceived quality  **(b)** Medium difference in perceived quality  **(c)** High difference in perceived quality

**Fig 8. Operating mode selection for simultaneous decision making by the platform.** ("—"indicates the platform profit in the com-mission-based model, "---" indicates the platform profit in the self-managed model).

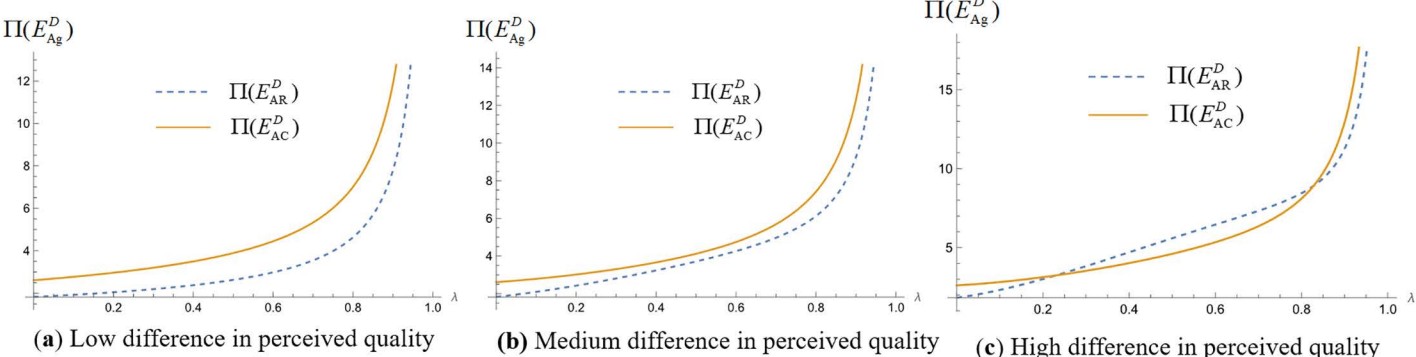

**(a)** Low difference in perceived quality  **(b)** Medium difference in perceived quality  **(c)** High difference in perceived quality

**Fig 9. Leader platform operation mode selection when the platform makes successive decisions.** ("—"indicates the platform profit in the commission-based model, "⋯" indicates the platform profit in the self-managed model).

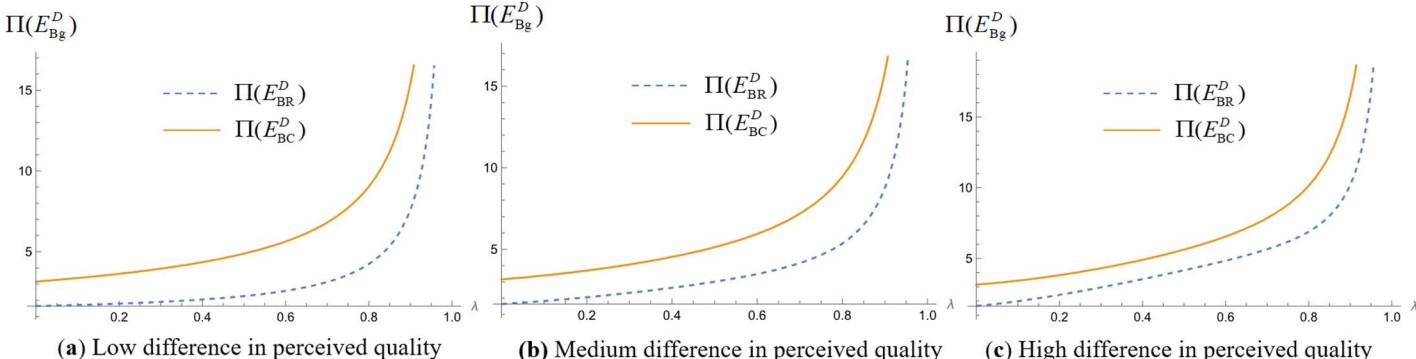

**(a)** Low difference in perceived quality  **(b)** Medium difference in perceived quality  **(c)** High difference in perceived quality

**Fig 10. Follower platform operation mode selection when the platform makes successive decisions.** ("—"indicates the platform profit in the commission-based model, "⋯" indicates the platform profit in the self-managed model).

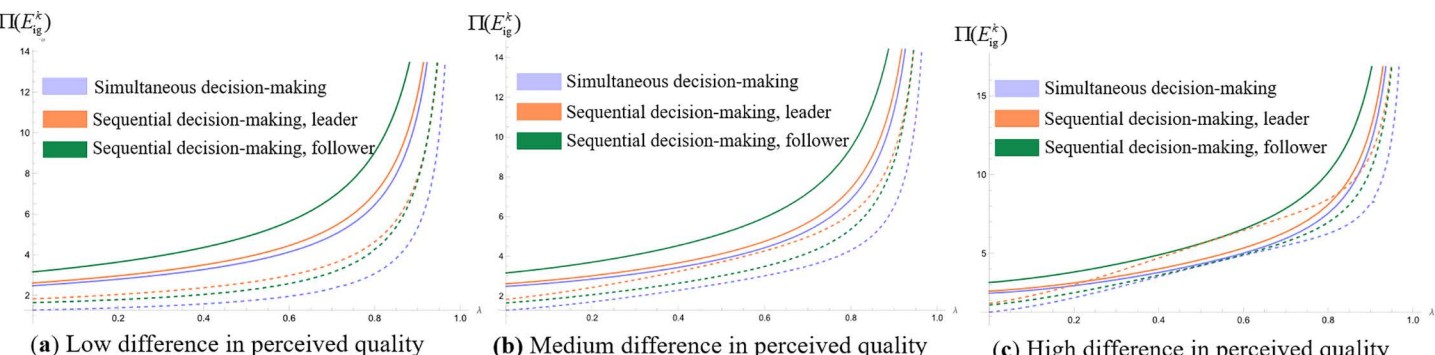

**(a)** Low difference in perceived quality  **(b)** Medium difference in perceived quality  **(c)** High difference in perceived quality

**Fig 11. Platform operation mode selection.** ("—"indicates the platform profit in the com-mission-based model, "⋯" indicates the platform profit in the self-managed model).

Fig 10(a), (b) and (c) represent the commission-based follower platform profit and self-owned follower platform profit when the two leader platforms make decisions successively. The perceived difference in product quality is small, the perceived difference in product quality is medium, and the perceived difference in product quality is large, respectively.

As illustrated in Fig 9, it can be seen that when the consumer product quality perception difference is small or at a medium level, the leader platform can obtain higher profit by choosing the commission-based model. When consumers' perception of product quality difference is large, the leader platform also needs to consider the probability of consumers getting a high-quality perception $\lambda$ when choosing operation mode: when the value of $\lambda$ is low or high, the leader platform can get higher platform profit by choosing commission mode; when $\lambda$ is at the medium level, the leader platform can get higher platform profit by choosing self-operated mode. This is because when the difference in consumer perception of product quality is small or at a medium level, the platform can reduce operating costs through the commission-based model, which helps the platform expand its business scope and market coverage and gain higher profit through advertising and commissions. When the difference in consumer product quality perception is large, the platform needs to provide high-quality products to attract consumers. Under the self-managed model, platforms have more direct control over product quality, which can better meet consumer demand for high-quality products and increase consumer satisfaction and loyalty. When the probability of consumers receiving high-quality perception is low or high, commission-based platforms can diversify risk by partnering with multiple manufacturers, taking advantage of competition among manufacturers to improve product quality and achieving higher profit at lower costs. When the probability of consumers obtaining high-quality perception is at a medium level, the captive platform can directly control product quality and better satisfy consumers' expectations for products with medium quality levels, resulting in higher profit. In addition, follower platforms cannot make decisions about which operating model to adopt and should be consistent with leader platforms.

As illustrated in Fig 10, it can be seen that when the leader platform makes decisions successively, the follower platform's platform profit is greater in the commission-based model than in the self-operated model, regardless of whether the difference in consumers' product quality perception is lower, higher, or at a medium level.

Figs 11(a), 11(b) and 11(c) represent the platform profits under the four operational strategies when the perceived difference in product quality is low, the perceived difference in product quality is medium and the perceived difference in product quality is high, respectively. The dashed line represents the platform profit under the self-operated model, while the solid line represents the platform profit under the commission-based model. Different colors are used to distinguish the order of platform decisions. Due to the symmetry of the best response functions of the two platforms in the 'simultaneous decision' strategy combination, the Nash equilibrium solution is unique and identical, so only six distinguishable profit curves are actually plotted in the figure. As can be seen from Fig 11, regardless of whether consumers' perceived differences in product quality are low, high, or at a medium level, when platforms make successive decisions, if a platform takes the leader role, it can achieve higher profits by choosing the self-operated operation model; if it takes the follower role, it can achieve higher profits by choosing the commission-based operation model. When platforms make simultaneous decisions, a platform can achieve higher profits by choosing the commission-based operation model.

This is because under the commission-based model, platform profits depend on merchant resources and commission rates, with no need to bear procurement or inventory costs. Leader platforms must invest resources in building traffic and rule systems, resulting in higher fixed costs; in contrast, followers can leverage existing markets and user habits to enter the market at low costs. By flexibly adjusting commission rates to seize merchant resources and avoid implicit costs, followers achieve better profits. Under the self-operated model, platform profits rely on scale effects and cost control capabilities. Leaders achieve high profits through a profit structure of "high sales volume + reasonable pricing" supported by their bargaining power in large-scale procurement, scale effects in logistics and warehousing, and brand trust. Followers, however, face issues such as high procurement costs, low logistics efficiency, and insufficient sales due to limited scale, falling into a "high cost - low profit" dilemma with profits lower than those of leaders. Under simultaneous decision-making, self-operated platforms tend to get caught in "price competition" to seize market share, and high fixed costs further

compress their profit margins. For commission-based platforms, whose profits come from merchant commissions, there is no need to directly participate in price wars. They can ensure profits by optimizing the quality of merchant resources and commission sharing ratios, making them less affected by aggressive competition.

## 7. Conclusions

Different from previous literature that focused on objective quality improvement or static price discrimination [28,31], this paper combines a two-period model to make consumer quality perception differences an endogenous core variable in two-period dynamic games, while also considering the interaction between platform operating models (self-operated/commission-based) and decision-making sequences (simultaneous/sequential). We respectively construct dynamic pricing game models under commission-based and self-operated models. Through comparative analysis and numerical simulation, this paper provides a theoretical basis for the platform to formulate effective operational strategies under different market factors. The main findings of this study are as follows: 1) Both in the commission-based (prices are decided by the manufacturers) model and the proprietary (prices are decided by the platforms) model, when the probability of consumers obtaining high-quality perception is high, the manufacturing enterprises and the platforms can increase the market price to obtain higher profit. 2) In the commission-based model, platforms can attract consumers by raising the commission rate and offering greater price concessions, thereby enhancing their profit and market competitiveness. 3) After locking in consumers with low prices in the first cycle, the leader platform can maintain its market share in the second cycle even if it raises the price for repeat purchases, thanks to its perceived advantage. This explains the sustainability of the "subsidize first, raise prices later" strategy in reality. 4)The widening perception gap has a positive effect on the profits of both platforms and manufacturers, and this conclusion holds true in both self-operated and commission-based models, indicating that differentiated experiences have become a new profit lever. 5) When platforms compete simultaneously, higher profit can be achieved by adopting a self-managed model. When competing platforms make successive decisions, if a platform takes the leader role, it can achieve higher profits by choosing the self-operated operation model; if it takes the follower role, it can achieve higher profits by choosing the commission-based operation model.

This study has important managerial insights for e-commerce platforms and manufacturers. Self-operated platforms should continue to invest in quality traceability systems and brand storytelling to amplify consumers' perception of high quality, thereby supporting subsequent price increases. Commission-based platforms can use high commission income to subsidize new users, attract low-perception migrants with coupons, and encourage manufacturers to disclose quality signals, creating a virtuous cycle. For manufacturers, when working with commission-based platforms, they can enhance consumer perception by strengthening digital quality control, thereby gaining higher commission tolerance and pricing flexibility from the platform. When working with self-operated platforms, they should focus on ensuring stable supply so that the platform can implement dynamic high-price strategies. This study is not only applicable to e-commerce platforms but also has significant implications for other industries such as retail and services that face multi-channel competition and consumer strategic behavior. Among these, the 3C electronics, beauty, and luxury goods categories stand out for their rapid iteration. In these industries, consumers have a fast-paced perception of updates and high price elasticity, and adopting the strategies outlined in this paper can quickly amplify profits.

This study has certain limitations. In the context of variations in consumer perception of product quality, this paper only considers the impact of price factors on channel competition. However, it is essential to acknowledge that buyer feedback [72] and return rates [73] also play significant roles in consumers' decisions regarding purchasing channels. Consequently, future research endeavors will delve deeper into online reviews and return policies to more precisely forecast market responses. Such research will comprehensively encapsulate the multifaceted factors influencing consumer perception of product quality, thereby affording enterprises more profound market insights and strategic directives.

## Supporting information

**S1 Appendix. Equilibrium solving.**
(DOCX)

## Author contributions

**Conceptualization:** Bo Xie.

**Data curation:** Bo Xie.

**Formal analysis:** Qiqi Guo.

**Investigation:** Muqing Niu.

**Methodology:** Bo Xie.

**Software:** Yingying Cheng.

**Supervision:** Qiqi Guo.

**Validation:** Bo Xie, Qiqi Guo.

**Visualization:** Yingying Cheng.

**Writing – original draft:** Bo Xie.

**Writing – review & editing:** Qiqi Guo, Muqing Niu.

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
