## [Decision Letter · Decision Letter 0]

28 Jul 2025

Dear Dr. Cheng,

Thank you for submitting your manuscript to PLOS ONE. After careful consideration, we feel that it has merit but does not fully meet PLOS ONE’s publication criteria as it currently stands. Therefore, we invite you to submit a revised version of the manuscript that addresses the points raised during the review process.

We look forward to receiving your revised manuscript.

Kind regards,

Prof. Biswajit Sarkar, Ph.D., Post-doc

Academic Editor

PLOS ONE

2. hank you for stating the following financial disclosure:

[This research was funded by Soft science project of Henan Province (grant No.:252400411280, 252400410046), Philosophy and Social Science Planning Project of Henan Province (grant No.: 2022CZH015), Philosophy and social science education strong provincial project of Henan Prov-ince (grant No.:2025JYQS1079), Henan Provincial Postgraduate Course Ideological and Political Demonstration Course Project (grant No.: YJS2024SZ14), National Social Science Foundation of China(grant No.:20BJL135),2024 funding project for international training of high-level talents in Henan Province: Research on the path and countermeasures of supply chain digital transfor-mation under the background of big data.].

Additional Editor Comments:

The paper is about “Two-period e-commerce platforms operation strategies considering the difference in product quality perception”.

I have the following comments:

• There is no big improvement in e-commerce platforms. How do you prove product quality perception?

• There are several equations. But how do you prove that those equations are correct? The derivation of each equation is needed in the revised version.

• How is the authenticity of the model confirmed? How is the data authenticated?

• What is the data source of the numerical experiment? Please note that the data is sourced from industry, literature, or any other relevant source, indicating whether it is accurate or artificial. Compare the results with the existing models.

• Avoid writing abbreviations in the abstract. Rewrite the abstract, mentioning the results obtained from your study. What research questions are you trying to solve in this model?

• What is the novelty of this study? Prove it by comparing it with the related newly introduced study. In the result section, put a subsection of discussions, where the main results obtained in this model are presented.

• Before the conclusion section, put the managerial sections and write how this study will benefit which type of industry and why. What are the limitations of studying? Put references in each future extension.

• The way of abstract writing is good, but it should contain the details of the study and the findings in a very constructive manner. The entire paper contains several grammatical and spelling errors in English. Could you correct them all?

• The research question is established in the introduction section. It should be rewritten. The introduction should be explained constructively but not in a lengthy manner. The introduction should be divided into paragraphs based on the research gap and the concept of the constructive model.

• The introduction must be broken into three parts: the necessity of the research, the research gap, and the research orientation. The literature review must be based on keywords.

• Keywords should be in the direction of the research. Write assumptions in more detail with references if available. Please write the significant findings in the conclusions. Do not mention all assumptions that have been indicated within the model.

Reviewers' comments:

Reviewer's Responses to Questions

**Comments to the Author**

1. Is the manuscript technically sound, and do the data support the conclusions?

Reviewer #1: Partly

Reviewer #2: Yes

2. Has the statistical analysis been performed appropriately and rigorously?

Reviewer #1: N/A

Reviewer #2: N/A

3. Have the authors made all data underlying the findings in their manuscript fully available?

Reviewer #1: No

Reviewer #2: Yes

4. Is the manuscript presented in an intelligible fashion and written in standard English?

Reviewer #1: No

Reviewer #2: Yes

Reviewer #1: Title: Two-period e-commerce platforms operation strategies considering the difference in product quality perception

Journal: PLOS ONE

Manuscript ID: PONE-D-25-31916

The study examines how consumer perceptions of product quality influence purchasing decisions on e-commerce platforms. The study has some merits in the present literature but there are some drawbacks which I have pointed out are given below.

1. Although the authors titled it "Two platform operation modes." (a) Self-employed model; (b) Commission-based model," however it appears to be reselling and agency selling modes in two parallel supply chains. If so, why is this type of name defined? Are authors attempted to demonstrate any particular situation that differed from the reselling and agency selling approaches?

2. Research questions are unclear. These should be highlighted more explicitly. Are your conclusions suitable for answering these questions?

3. Figure 1 is unclear and difficult to understand. Please redraw it to demonstrate the clarity of your proposed model.

4. The authors claim that two supply chains made choices simultaneously or sequentially. But how do the two players' decisions flow in each supply chain? Are they playing the Nash and Stackelberg game? Nothing is clearly stated in this study. Also, what about the two-period decision sequence? The research provides no obvious answers.

5. There is no analysis that compares all four scenarios. Provide some analyses that contain all four models' optimal outcomes.

6. Provide a table comparing the optimal numerical results for all scenarios.

7. Why are Fig. 4(a) and (b) plotted in two parts? It would be preferable to plot both scenarios together to compare their variations. Similar comments apply to Figs. 5(a-d).

8. The placement of figures and titles can be confusing for readers. Provide both the title and the figure in the same place, either within the text or at the end of the text.

9. The authors' major conclusions highlight the study's important findings. However, those require a more sophisticated presentation.

10. Add management implications in the conclusion section.

11. Provide references to elaborate on the study's limitations and future directions.

12. The figures lack emphasis and clarity. Please redo these in a flawless manner.

The paper must be significantly improved in view of these issues before it can be given consideration for probable publication in PLOS ONE.

Reviewer #2: Two-period e-commerce platforms operation strategies considering the difference in product quality perception

This study examines a product retail supply chain consisting of two e-commerce platforms, two manufacturers, and strategic consumers. The authors construct a two-period dynamic pricing game model to compare commission-based and self-operated models for the platforms. Based on different operating modes and decision sequences, four scenarios are considered. The research is relatively comprehensive and carries both theoretical significance and practical relevance. However, several key issues remain to be addressed. Therefore, a “Minor Revision” is recommended. The specific suggestions are as follows:

1. The statement “we provide a new theoretical framework for…” does not sufficiently demonstrate how the study fills the theoretical gap. The authors should clearly articulate the innovations and contributions in the Introduction to enhance the paper’s theoretical value.

2. The literature review section requires appropriate adjustment to strengthen its alignment with the focus of the paper. In particular, the discussion should emphasize the three highly relevant streams of literature: strategic consumers (including purchase intention, quality perception, and price discrimination), operational models of e-commerce platforms, and decision-making sequences. A clearer integration of these themes would improve the coherence and academic contribution of the review.

3. Figure 1 should be revised, as the differences between the self-operated and commission-based models are minimal. Additionally, since the paper employs the Hotelling model, a brief explanation should be provided to improve clarity and readability.

4. Several formatting and consistency issues should be corrected, such as inconsistent expressions like “4.1. Self-reliant model” and the presence of Chinese symbols in the formulas of Theorem 5 and Theorem 6.

5. In Section 6: Numerical Simulation, the sources of numerical values are not specified, which reduces the practical relevance of the study. Furthermore, when comparing multiple figures, most results remain unchanged, making the analysis appear somewhat repetitive.

**Do you want your identity to be public for this peer review?** For information about this choice, including consent withdrawal, please see our For information about this choice, including consent withdrawal, please see our Privacy Policy .

Reviewer #1: No

Reviewer #2: No

While revising your submission, please upload your figure files to the Preflight Analysis and Conversion Engine (PACE) digital diagnostic tool, https://pacev2.apexcovantage.com/ . PACE helps ensure that figures meet PLOS requirements. To use PACE, you must first register as a user. Registration is free. Then, login and navigate to the UPLOAD tab, where you will find detailed instructions on how to use the tool. If you encounter any issues or have any questions when using PACE, please email PLOS at . PACE helps ensure that figures meet PLOS requirements. To use PACE, you must first register as a user. Registration is free. Then, login and navigate to the UPLOAD tab, where you will find detailed instructions on how to use the tool. If you encounter any issues or have any questions when using PACE, please email PLOS at figures@plos.org . Please note that Supporting Information files do not need this step.. Please note that Supporting Information files do not need this step.

---

## [Author Response · Author response to Decision Letter 1]

24 Aug 2025

Thanks again for your consideration of publishing our manuscript in your journal. We have conducted a comprehensive revision of the entire manuscript. In response to reviewers, the reviewer’s comments are presented in italica, and our corresponding changes and additions to the manuscript are highlighted in red text. We have tried our best to make all revisions clear, and we hope that the revised manuscript meets the requirements for publication.

---

## [Decision Letter · Decision Letter 1]

24 Oct 2025

Dear Dr. Cheng,

Thank you for submitting your manuscript to PLOS ONE. After careful consideration, we feel that it has merit but does not fully meet PLOS ONE’s publication criteria as it currently stands. Therefore, we invite you to submit a revised version of the manuscript that addresses the points raised during the review process.

We look forward to receiving your revised manuscript.

Kind regards,

Prof. Biswajit Sarkar, Ph.D., Post-doc

Academic Editor

PLOS ONE

Journal Requirements:

Additional Editor Comments:

The paper must be revised based on the reviewers' comments, and the significant comment from the Editor is that it must be compared with existing research to demonstrate the novelty of this study. The comparison must be based on theoretical, numerical, and industrial perspectives.

The author should carefully revise the paper, as if the author will revise it very carefully, to prove the novelty by comparing it with existing articles. I may recommend acceptance of the paper for publication in the next revision.

Reviewer's Responses to Questions

**Comments to the Author**

Reviewer #1: All comments have been addressed

Reviewer #2: All comments have been addressed

Reviewer #3: (No Response)

2. Is the manuscript technically sound, and do the data support the conclusions?

Reviewer #1: Yes

Reviewer #2: Partly

Reviewer #3: Yes

3. Has the statistical analysis been performed appropriately and rigorously?

Reviewer #1: Yes

Reviewer #2: N/A

Reviewer #3: Yes

4. Have the authors made all data underlying the findings in their manuscript fully available?

Reviewer #1: Yes

Reviewer #2: Yes

Reviewer #3: No

5. Is the manuscript presented in an intelligible fashion and written in standard English?

Reviewer #1: Yes

Reviewer #2: Yes

Reviewer #3: Yes

Reviewer #1: Authors answered all my previous comments very carefully. So, the paper can be accepted after these minor changes:

1. Paper is very lengthy, if possible, delete less important or repeated topics.

2. In my previous comment “Provide a table comparing the optimal numerical results for all scenarios.” Which is placed in Table 3 in the revised manuscript. In this comment, I want to show the results numerical values not expressions.

3. Conclusion section must be divided into main findings, managerial insights and limitations with future research directions.

Reviewer #2: The author has made substantial revisions in response to the suggestions provided, and most of the significant issues have been carefully addressed. Therefore, I recommend to Accept the manuscript.

Reviewer #3: This manuscript addresses the topic of operational strategies for e-commerce platforms in the presence of differences in consumer product quality perception. The study develops a two-period dynamic pricing game model that incorporates two platforms, two manufacturers, and strategic consumers. The authors compare the commission-based model and the self-operated model under both simultaneous and sequential decision-making, leading to four distinct scenarios (S-R, S-C, D-R, D-C). The results demonstrate how consumer quality perception probabilities influence optimal pricing strategies, platform profits, and the choice of operation modes across different decision structures. For example, the paper shows that when platforms make decisions simultaneously, the self-operated model is preferable, whereas under sequential decision-making, the optimal choice depends on both perceived product quality differences and consumer perception probabilities. Importantly, the paper provides insights into how platforms can attract consumers with low prices in the first period and retain them even with higher prices in the second period.

Overall, this is a well-structured and carefully executed study. Considering that previous reviewers have already raised most of the essential methodological and conceptual questions, and that the current version adequately addresses the key issues, I find the manuscript ready for publication.

**Do you want your identity to be public for this peer review?** For information about this choice, including consent withdrawal, please see our For information about this choice, including consent withdrawal, please see our Privacy Policy .

Reviewer #1: No

Reviewer #2: No

Reviewer #3: No

---

## [Author Response · Author response to Decision Letter 2]

26 Nov 2025

Thank you for your valuable feedback. We have removed our figures from within our manuscript file, leaving only the individual TIFF/EPS image files. Furthermore, all relevant data are within the manuscript and its Supporting Information files and we confirm that our submission contains our "minimal data set". We hope that the revised manuscript meets the publication requirements.

---

## [Decision Letter · Decision Letter 2]

5 Mar 2026

Two-period e-commerce platforms operation strategies considering the difference in product quality perception

PONE-D-25-31916R2

Dear Dr. Cheng,

We’re pleased to inform you that your manuscript has been judged scientifically suitable for publication and will be formally accepted for publication once it meets all outstanding technical requirements.

Kind regards,

Manuel Herrador, Ph.D.

Academic Editor

PLOS One

Additional Editor Comments (optional):

Dear authors,

I am pleased to inform you that your manuscript has been accepted for publication in PLOS ONE.

Both reviewers were satisfied with your revisions and have recommended acceptance. Your study is rigorous and makes a valuable addition to the literature.

Thank you for choosing to submit your research to PLOS ONE. The editorial office will contact you shortly regarding the next steps for production and proofing.

Best regards

Reviewers' comments:

Reviewer's Responses to Questions

**Comments to the Author**

Reviewer #1: All comments have been addressed

Reviewer #2: All comments have been addressed

2. Is the manuscript technically sound, and do the data support the conclusions?

Reviewer #1: Yes

Reviewer #2: (No Response)

3. Has the statistical analysis been performed appropriately and rigorously?

Reviewer #1: Yes

Reviewer #2: (No Response)

4. Have the authors made all data underlying the findings in their manuscript fully available?

Reviewer #1: Yes

Reviewer #2: (No Response)

5. Is the manuscript presented in an intelligible fashion and written in standard English?

Reviewer #1: Yes

Reviewer #2: (No Response)

Reviewer #1: The author has made substantial revisions in response to my previous comments. The most of the significant issues have been carefully addressed. Therefore, I recommend to Accept the manuscript.

Reviewer #2: The authors have meticulously addressed the concerns raised in the previous round of review. Most of the critical issues have been successfully resolved or adequately clarified through the revisions. The manuscript has been significantly improved and now meets the high standards of the journal. Therefore, I recommend the manuscript for publication with minor revisions.

**Do you want your identity to be public for this peer review?** For information about this choice, including consent withdrawal, please see our For information about this choice, including consent withdrawal, please see our Privacy Policy .

Reviewer #1: No

Reviewer #2: **Yes:** Weisi ZhangWeisi Zhang

---

## [Editor Report · Acceptance letter]

PONE-D-25-31916R2

PLOS One

Dear Dr. Cheng,

I'm pleased to inform you that your manuscript has been deemed suitable for publication in PLOS One. Congratulations! Your manuscript is now being handed over to our production team.

Kind regards,

on behalf of

Dr. Manuel Herrador

Academic Editor

PLOS One